# Soluble C-Type Lectin-Like Receptor 2 Elevation in Patients with Acute Cerebral Infarction

**DOI:** 10.3390/jcm10153408

**Published:** 2021-07-30

**Authors:** Akisato Nishigaki, Yuhuko Ichikawa, Minoru Ezaki, Akitaka Yamamoto, Kenji Suzuki, Kei Tachibana, Toshitaka Kamon, Shotaro Horie, Jun Masuda, Katsutoshi Makino, Katsuya Shiraki, Hideto Shimpo, Motomu Shimaoka, Katsue Suzuki-Inoue, Hideo Wada

**Affiliations:** 1Department of Neurology, Mie Prefectural General Medical Center, Yokkaichi 510-8561, Japan; nishigaki-a@med.mie-u.ac.jp (A.N.); ksuzu6161@gmail.com (K.S.); ak1233maggot@gmail.com (K.T.); toshi9031@yahoo.co.jp (T.K.); holly.0639-bj@i.softbank.jp (S.H.); 2Department of Central Laboratory, Mie Prefectural General Medical Center, Yokkaichi 510-8561, Japan; ichi911239@yahoo.co.jp (Y.I.); ajbyd06188@yahoo.co.jp (M.E.); 3Department of Emergency and Critical Care Center, Mie Prefectural General Medical Center, Yokkaichi 510-8561, Japan; akitaka-yamamoto@mie-gmc.jp; 4Department of Cardiovascular Medicine, Mie Prefectural General Medical Center, Yokkaichi 510-8561, Japan; jun-masuda@mie-gmc.jp (J.M.); katsutoshi-makino@mie-gmc.jp (K.M.); 5Department of Laboratory and General Medicine, Mie Prefectural General Medical Center, Yokkaichi 510-8561, Japan; katsuya-shiraki@mie-gmc.jp; 6Mie Prefectural General Medical Center, Yokkaichi 510-8561, Japan; hideto-shimpo@mie-gmc.jp; 7Department of Molecular Pathobiology and Cell Adhesion Biology, Mie University Graduate School of Medicine, Tsu 514-8507, Japan; motomushimaoka@gmail.com; 8Department of Clinical and Laboratory Medicine, University of Yamanashi, Yamanashi 409-3898, Japan; katsuei@yamanashi.ac.jp

**Keywords:** platelet activation, sCLEC-2, acute cerebral infarction (ACI), atherosclerotic ACI, cardioembolic ACI

## Abstract

Background: Acute cerebral infarction (ACI) includes cardiogenic ACI treated with anticoagulants and atherosclerotic ACI treated with antiplatelet agents. The differential diagnosis between cardiogenic and atherosclerotic ACI is still difficult. Materials and Methods: The plasma sCLEC-2 and D-dimer levels were measured using the STACIA system. Results: The plasma sCLEC-2 level was significantly high in patients with ACI, especially those in patients with atherosclerotic or lacunar ACI, and plasma D-dimer levels were significantly high in patients with cardioembolic ACI. The plasma levels of sCLEC-2 and the sCLEC-2/D-dimer ratios in patients with atherosclerotic or lacunar ACI were significantly higher than those in patients with cardioembolic ACI. The plasma D-dimer levels in patients with atherosclerotic or lacunar ACI were significantly lower than those in patients with cardioembolic ACI. The plasma levels of sCLEC-2 and the sCLEC-2/D-dimer ratios were significantly higher in patients with atherosclerotic or lacunar ACI or acute myocardial infarction in comparison to patients with cardioembolic ACI or those with deep vein thrombosis. Conclusion: Using both the plasma sCLEC-2 and D-dimer levels may be useful for the diagnosis of ACI, and differentiating between atherosclerotic and cardioembolic ACI.

## 1. Introduction

Stroke occurs in more than 13.7 million people each year, with 5.8 million people dying each year as a consequence. Approximately 70% (9.5 million people) of incident strokes are ischemic [1,2]. An ischemic stroke, acute cerebral infarction (ACI), a type of an ischemic stroke, is classified into following entities: cardioembolic ACI [3], atherosclerotic ACI [4], and lacunar ACI [5]. As the patients with ACI still have a high mortality, they require prompt treatment. Computed tomography (CT), magnetic resonance imaging (MRI) and angiography are required for the early diagnosis for ACI, and the differential diagnosis between cardioembolic ACI and atherosclerotic or lacunar ACI may be difficult. However, a patient with cardioembolic ACI is treated with vitamin K antagonist or direct oral anticoagulant, while a patient with atherosclerotic or lacunar ACI or transient ischemic attack (TIA) is treated with aspirin or other antiplatelet agents [1,6], because, it is considered that the platelet activation may play an important role in atherosclerosis [7]. There are few useful biomarkers for the detection of ACI or which can be applied in the differential diagnosis of atherosclerotic ACI and cardioembolic ACI.

The biomarkers for platelet activation may be a platelet factor 4 (PF4), β-thromboglobulin (β-TG), and P-selectin; however their actual diagnostic specificity for thrombosis due to platelet activation is not high, and they have limited use in the clinical laboratory. C-type lectin-like receptor 2 (CLEC-2) is a platelet activation receptor for the membrane protein podoplanin, which is expressed on some types of tumor cells and lymphatic endothelial cells. CLEC-2/podoplanin interaction facilitates tumor metastasis and blood/lymphatic vessel separation and normal lung formation during embryonic development [8,9]. Soluble C-type lectin-like receptor 2 (sCLEC-2) is released upon platelet activation [10] and it has been introduced as new biomarker of platelet activation. Soluble C-type lectin-like receptor 2 (sCLEC-2) has been thus introduced as new biomarkers of platelet activation [11,12]. Elevated plasma levels of sCLEC-2 were reported in patient with thrombotic microangiopathy (TMA) and disseminated intravascular coagulation (DIC) [12], as well as patients with acute coronary syndrome [13,14] and acute ischemic stroke [15].

In the present study, the plasma sCLEC-2 and D-dimer levels were measured in 77 patients with ACI and 254 patients with other diseases to examine its diagnostic value in ACI.

## 2. Materials and Methods

The plasma sCLEC-2 and D-dimer levels were measured in 331 critical ill patients at Mie Prefectural General Medical Center from 1 August 2019 to 30 April 2020. There were 77 patients with CI, 11 with pulmonary embolism (PE), 12 with deep vein thrombosis (DVT), 7 with TIA, 26 with acute myocardial infarction (AMI), 7 with syncope, 28 with digestive system disease, 52 with heart failure, 30 with benign anemia or idiopathic thrombocytopenic purpura (ITP), 75 with indefinite compliant syndrome (UCS) (Table 1). The patients with ACI include 18 patients with cardioembolic ACI, 36 patients with atherosclerotic ACI, and 23 patients with lacunar ACI. Blood samples (1.8 mL) were obtained in plastic tube containing 0.2 mL of 3.2% sodium citrate buffer at admission and before antiplatelet or anticoagulant therapy. Plasma was obtained after the centrifugation of blood sample two times (3000× *g*).

The study protocol (O-0057) was approved by the Human Ethics Review committees of Mie Prefectural General Medical Center, and informed consent was obtained from each patient.

### 2.1. Measurement of the sCLEC-2, and D-Dimer Levels

We measured the plasma sCLEC-2 levels by a chemiluminescent enzyme immunoassay (CLEIA) using previously described monoclonal antibodies and the STACIA CLEIA system (LSI Medience, Tokyo, Japan). In brief, magnetic particles were coated with the anti-sCLEC-2 monoclonal antibody 5D11. The plasma samples were then incubated with antibody-coated magnetic particles, and after washing, they were incubated further with the alkaline-phosphatase-labeled anti-sCLEC-2 monoclonal antibody 11E6. After washing again, the magnetic particles were incubated with chemiluminescent substrate solution (CDP-Star; Applied BioSystems) and the luminescence was measured using the luminometer installed in the STACIA system. The D-dimer levels were determined via the latex agglutination method using LPIA-GENESIS D-Dimer (LSI Medience).

### 2.2. Statistical Analyses

The data are expressed as the median (25th to 75th percentile). Differences between groups were examined for significance using the Mann-Whitney U test. *p*-values of ≤0.05 were considered to indicate statistical significance. All statistical analyses were performed using the StatFlex software program (version 6: Artec Co. Ltd., Osaka, Japan).

## 3. Results

The age, sex, platelet count, APTT, PT-INR, and D-dimer levels in patients with cerebral infarction (CI), pulmonary embolism (PE), deep vein thrombosis (DVT), transient ischemic attack (TIA), acute myocardial infarction (AMI), syncope, digestive system disease, heart failure, benign anemia or idiopathic thrombocytopenia purpura (ITP), or indefinite compliant syndrome (UCS) are shown in Table 1. The age was higher in patients with heart failure, the platelet count was lower in patients with heart failure or benign anemia and ITP, the PT-INR was higher in patients with benign anemia and ITP, and the D-dimer level was higher in patients with PE, DVT, or heart failure in comparison with patients with ACI. The plasma sCLEC-2 (mean ± standard deviation) level in 79 healthy volunteers was 59.1 ± 16.7 pg/mL. The plasma sCLEC-2 (median; 25–75 percentile) levels of patients with ACI (256 pg/mL; 182–340 pg/mL) were significantly higher in comparison to those in patients with DVT (176 pg/mL; 128–188 pg/mL), syncope (156 pg/mL; 121–182 pg/mL), digestive system disease (209 pg/mL; 139–302 pg/mL), heart failure (230 pg/mL; 171–287 pg/mL), benign anemia or ITP (156 pg/mL; 104–197 pg/mL) and UCS (192 pg/mL; 143–241 pg/mL) (Figure 1). The relationship between plasma sCLEC-2 levels (X) and platelet count (Y) was Y = 111 + 5.804 X (r = 0.349, *p* < 0.001).

The plasma sCLEC-2 levels of patients with atherosclerotic ACI (276 pg/mL; 220–381 pg/mL) were significantly higher in comparison to those of patients with cardioembolic ACI (208 pg/mL; 158–253 pg/mL) or syncope (156 pg/mL; 121–182 pg/mL). Furthermore, the plasma sCLEC-2 levels of patients with lacunar ACI (258 pg/mL; 190–309 pg/mL) were significantly higher in comparison to patients with syncope (Figure 2a). The plasma D-dimer levels of patients with cardioembolic ACI (2.4 μg/mL; 0.8–5.0 μg/mL) were significantly higher than those in patients with atherosclerotic ACI (1.0 μg/mL; 0.6–1.9 μg/mL), those with lacunar ACI (0.7 μg/mL; 0.4–1.3 μg/mL), and patients with TIA (0.7 μg/mL; 0.6–0.9 μg/mL) (Figure 2b). The sCLEC-2/D-dimer ratios of patients with atherosclerotic ACI or lacunar ACI were significantly higher in comparison to those in patients with cardioembolic ACI or syncope (Figure 2c). The plasma sCLEC-2 levels of patients with atherosclerotic or lacunar ACI were significantly higher in comparison to those of patients with cardioembolic ACI (*p* < 0.05). The plasma D-dimer levels in patients with atherosclerotic or lacunar ACI were significantly lower in comparison to those in patients with cardioembolic ACI (*p* < 0.01). Furthermore, the plasma D-dimer or sCLEC-2 levels of patients with atherosclerotic or lacunar ACI or AMI were significantly different in comparison to those in patients with cardioembolic ACI or DVT (*p* < 0.001 or *p* < 0.01, respectively) (Figure 3a,b). The sCLEC-2/D-dimer ratios sufficiently enhanced the difference not only between atherosclerotic or lacuna ACI and cardioembolic ACI but also enhanced the difference between venous thromboses (e.g., cardioembolic ACI and VTE) and atherosclerotic thromboses (e.g., lacuna and atherosclerotic ACI and AMI) (Figure 3c).

## 4. Discussion

ACI occurs when thrombosis or embolism occludes a cerebral vessel supplying a specific area of the brain resulting in a loss of neurological function [1]. Thus, an early diagnosis and treatment are required in the management of ACI. D-dimer is a sensitive but not spesific marker for diagnosing PE/DVT [16] and DIC [17]. Although plasma D-dimer levels are slightly high in patients with cardioembolic ACI, they were low in patients with atherosclerotic or lacunar ACI in this study. Meanwhile, the plasma sCLEC-2 levels were markedly high in patients with ACI including those with atherosclerotic or lacunar ACI. The high plasma sCLEC-2 levels in patients with lacunar ACI (a small infarction) or TIA, suggest that sCLEC-2 has a high sensitivity in the diagnosis of an ischemic attack. The plasma sCLEC-2 levels were markedly high in patients with TMA, DIC, and acute coronary syndrome [12,13,14], suggesting that elevated sCLEC-2 levels might reflect platelet activation and that they may not directly reflect infarction. As the plasma sCLEC-2 levels were high in patients with lacunar ACI or TIA, these levels might be high in patients with non-symptomatic ACI [18].

CLEC-2 protein is highly and almost specifically expressed in platelets and megakaryocytes in humans, but it is also expressed at lower levels in the liver Kupffer cells [8]. PF4 andβ-TG are contained in α-granule and are used as classical platelet activation markers, while CLEC-2 is a platelet membrane protein. sCLEC-2 has several advantages over the classical platelet activation markers. PF4 and β-TG are easily released upon the minimal platelet activation that occurs during sampling and require plasma mixed with citrate, adenosine, theophylline and adenosine (CTAD); however, the effects of anti-coagulants on the sCLEC-2 ELISA were negligible. Moreover, while special techniques are required for blood sampling and sample preparation for PF4 and β-TG measurement, the standard blood collection procedures used in daily clinical laboratory tests have shown to be sufficent for sCLEC-2 measurement [10]. sCLEC-2 is derived via membrane shedding and microvesicles. We have shown that in the ELISA system to measure sCLEC-2, both forms of sCLEC-2 are detected using 5D11 and 11E6 [10]. Since the CLEIA uses the same set of monoclonal anti-CLEC-2 antibodies, we assume that the measured sCLEC-2 is derived both via membrane shedding and from microvesicles.

The mechanisms underlying the onset of ACI differ between atherosclerotic and cardioembolic ACI [19] and the managements of these types of ACI are also different. That is, cardioembolic ACI patients were mainly treated with anticoagulant therapy and atherosclerotic ACI patients were mainly treated with antiplatelet therapy. Although the differential diagnosis between atherosclerotic and cardioembolic ACI is required, it can be difficult to differentiate between these two types of ACI by CT or MRI. As the plasma sCLEC2 levels are high in atherosclerotic ACI and the plasma D-dimer levels are high in cardioembolic ACI, the sCLEC-2/D-dimer ratio was evaluated. This ratio sufficiently enhanced the difference not only between atherosclerotic and cardioembolic ACI but also the difference between venous thromboses (e.g., cardioembolic ACI and VTE) and atherosclerotic thromboses (e.g., lacuna and atherosclerotic ACI and AMI), suggesting that elevated sCLEC-2/D-dimer ratio was stronger adaptive indicator for antiplatelet therapy than only elevated sCLEC-2.

## 5. Conclusions

sCLEC-2 levels were significantly increased in patients with ACI, and using both the plasma sCLEC-2 and D-dimer levels may be useful for the diagnosis of ACI, and differentiating between atherosclerotic and cardioembolic ACI.

## 6. Patent

Hideo Wada and Katsue Suzuki-Inoue are inventors of the patent application related to this report. (Japanese Patent Application No. 2021-091606 and 6078845, respectively).

## Figures and Tables

**Figure 1 jcm-10-03408-f001:**
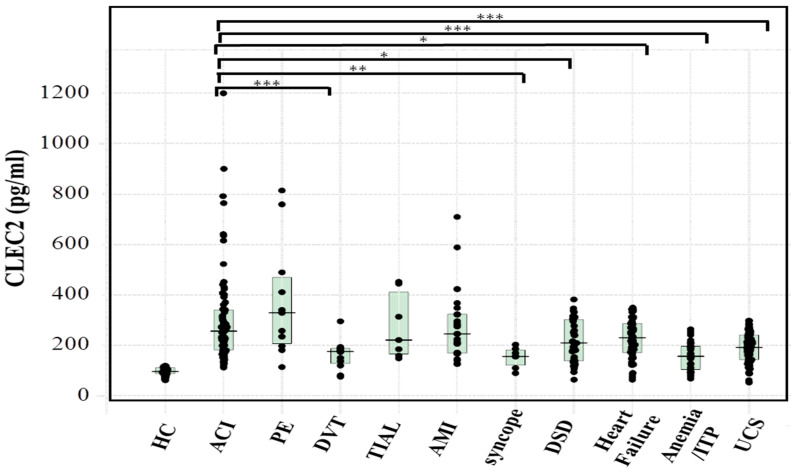
The plasma sCLEC-2 levels in various diseases. HC (*n* = 70), healthy control; ACI, acute cerebral infarction; PE, pulmonary embolism; DVT, deep vein thrombosis; TIAL, transient ischemic symptoms for cerebral infarction within 24 h; AMI, acute myocardial infarction; DSD, digestive system disease; ITP, idiopathic thrombocytopenic purpura; UCS, indefinite compliant syndrome; *** *p* < 0.001; ** *p* < 0,01; * *p* < 0.05.

**Figure 2 jcm-10-03408-f002:**
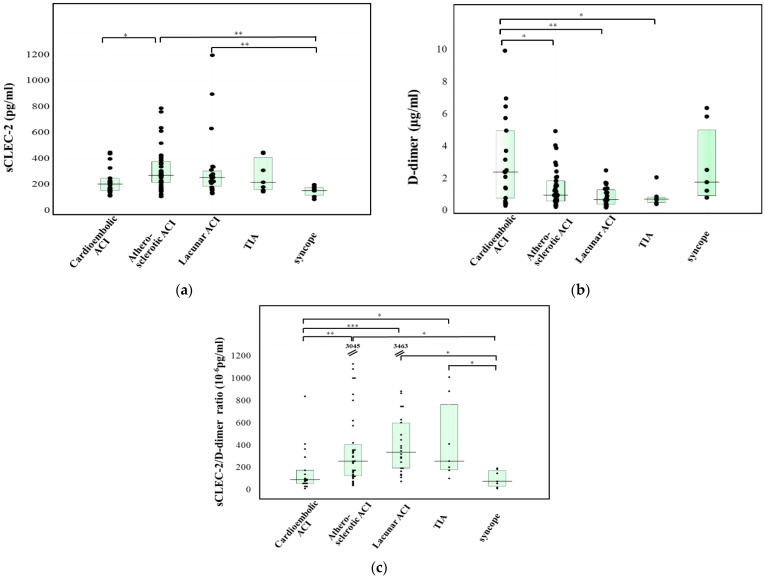
The plasma levels of sCLEC-2 (**a**), and D-dimer (**b**), and the CLEC-2/D-dimer ratio (**c**) in patients with cardioembolic ACI, with atherosclerotic ACI, with Lacunar CI, with TIA, or with syncope. ACI, acute cerebral infarction; *** *p* < 0.001; ** *p* < 0.01; * *p* < 0.05; The numbers in (**c**) indicated sCLEC-2/D-dimer ratio ≥1200.

**Figure 3 jcm-10-03408-f003:**
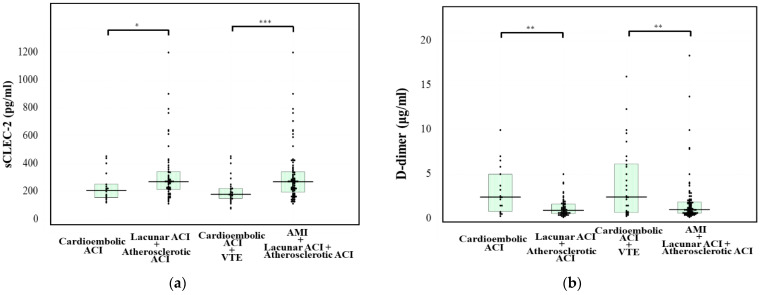
The plasma levels of sCLEC-2 (**a**), and D-dimer (**b**), and the sCLEC-2/D-dimer ratio (**c**) in patients with cardioembolic ACI, with atherosclerotic ACI and lacunar ACI, with cardioembolic ACI and VTE, or with AMI and atherosclerotic ACI and lacunar ACI. ACI, acute cerebral infarction; VTE, venous thromboembolism; AMI, acute myocardial infarction; *** *p* < 0.001; ** *p* < 0.01; * *p* < 0.05.

**Table 1 jcm-10-03408-t001:** Subjects.

	Age (Years)	Sex (F:M)	Platelets (×10^9^/L)	APTT (s)	PT-INR	D-Dimer (μg/mL)
ACI	77.0 (60.5–81.8)	26:51	227 (186–269)	29.0 (27.0–32.0)	0.97 (0.94–1.02)	1.1 (0.6–2.1)
Cardioembolic ACI	81.5 * (71.0–90.0)	5:13	208 (137–254)	31.0 * (29.8–33.3)	1.01 ** (0.98–1.08)	2.6 * (0.9–4.6)
Atherosclerotic ACI	72.0 (67.0–78.5)	12:24	241 (185–271)	29.0 (27.0–30.0)	0.96 (0.93–1.00)	1.0 (0.5–2.0)
Lacunar ACI	72.0 (64.3–78.8)	9:14	226 (192–268)	29.0 (26.0–33.0)	0.95 (0.89–0.98)	0.8 (0.5–1.4)
Pulmonary embolism	66.0 (48.0–74.5)	8:3	241 (146–279)	34.0 (0.3–48.5)	1.11 *** (1.08–1.37)	5.9 *** (4.2–17.6)
Deep vein thrombosis	69.0 (64.8–74.0)	7:5	209 (191–253)	31.0 (28.0–34.5)	1.00 (0.95–1.08)	5.1 * (0.7–11.0)
TIA	79.0 (70.0–87.0)	3:4	191 (184–321)	28.0 (25.5–29.8)	0.95 ** (0.91–1.05)	0.7 (0.6–0.9)
AMI	79.0 (70.0–87.0)	10:16	190 (154–261)	31.5 (27.0–54.0)	1.02 (0.96–1.23)	1.7 (0.9–3.1)
Syncope	76.0 (36.5–83.3)	4:3	207 (144–255)	30.0 (27.3–30.0)	0.99 (0.93–1.08)	1.8 (1.0–5.1)
DSD	67.5 (54.0–80.0)	15:23	227 (198–277)	28.0 (27.0–32.0)	1.02 (0.93–1.08)	0.9 (0.4–1.6)
Heart failure	81.0 * (74.0–86.5)	22:26	177 *** (131–218)	31.0 (28.0–34.8)	1.11 *** (1.02–1.24)	2.8 *** (1.4–5.4)
Anemia/ITP	70.0 (62.0–80.0)	20:10	122 *** (55–186)	31.0 * (29.0–39.0)	1.04 ** (1.01–1.07)	0.5 * (0.4–1.2)
UCS	57.0 (48.0–73.0)	40:35	234 (190–275)	29.0 (27.0–31.8)	0.96 (0.91–1.00)	0.6 ** (0.4–1.5)

Data are expressed as the median (25–75 percentile). ACI, acute cerebral infarction; TIA, transient ischemic attack, AMI, acute myocardial infarction; DSD. Digestive system disease; ITP, idiopathic thrombocytopenic purpura; UCS, indefinite compliant syndrome; *** *p* < 0.001; ** *p* < 0.01; * *p* < 0.05 in comparison to patients with ACI; F, female; M, male; APTT, activated partial thromboplastin time; PT-INR, prothrombin time-international normalized ratio.

## Data Availability

The data presented in this study are available on request to the corresponding author. The data are not publicly available due to privacy restrictions.

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
