# Peer review of "Soluble C-Type Lectin-Like Receptor 2 Elevation in Patients with Acute Cerebral Infarction"

_jcm, 2021, doi:10.3390/jcm10153408_

Round 1

Reviewer 1 Report

Authors have written an interesting article about important topic. However, there are few things to be considered.

Introduction

Please provide some basic information (1-2 sentences) about soluble C-type lectin- 61 like receptor 2 (sCLEC-2); "What is it´s normal biological function in body?" etc.

Lines 65-67: "Soluble platelet membrane glycoprotein VI (sGPVI) is a similar soluble platelet membrane protein to sCLEC-2, and is also reported to be high in patients with DIC, TMA, and acute coronary syndrome and post-operation patients [12-14]." This information seems irrelevant to this article. Please remove it.

Results

Line 147: Some extra space in the end of line.

Discussion

Line 158: Extra space between words "area" and "of".

Line 159: Cannot agree that D-dimer (alone) is useful in clinical practise. It is sensitive. But it is not spesific. Please correct the sentence. For example; "D-dimer is sensitive but not spesific marker for diagnosing PE/DVT and DIC". There are also more accurate references for this topic. For example; https://www.sciencedirect.com/science/article/pii/S0735109717397590?via%3Dihub

Author Response

We thank you for your helpful and valuable comments on our study titled, “Soluble C-type lectin-like receptor 2 elevation in patients with acute cerebral infarction.” (Ref.: Manuscript Number: jcm-1305814). We have responded to all suggestions from the reviewers and have revised the manuscript accordingly. Our corrections are indicated as red letters with yellow highlight in the revised manuscript.

Respectfully yours,

Hideo Wada

 Introduction

Comment 1

Please provide some basic information (1-2 sentences) about soluble C-type lectin- 61 like receptor 2 (sCLEC-2); "What is it´s normal biological function in body?" etc.

Response 1 We have added the following description to the Introduction.             “C-type lectin-like receptor 2 (CLEC-2) is a platelet activation receptor for the membrane protein podoplanin, which is expressed on some types of tumor cells and lymphatic endothelial cells. CLEC-2/podoplanin interaction facilitates tumor metastasis and blood/lymphatic vessel separation and normal lung formation during embryonic development [8, 9]. Soluble C-type lectin-like receptor 2 (sCLEC-2) is released upon platelet activation [10] and it has been introduced as a new biomarker of platelet activation.”  

Comment 2

Lines 65-67: "Soluble platelet membrane glycoprotein VI (sGPVI) is a similar soluble platelet membrane protein to sCLEC-2, and is also reported to be high in patients with DIC, TMA, and acute coronary syndrome and post-operation patients [12-14]." This information seems irrelevant to this article. Please remove it.

Response 2 This sentence and references 12-14 were removed in accordance with the reviewer’s comments.

 Results

 Comment 3

Line 147: Some extra space in the end of line.

Response 3 The extra spaces were corrected.

 Discussion

 Comment 4

Line 158: Extra space between words "area" and "of".

Response 4 The extra space was corrected.

 Comment 5

Line 159: Cannot agree that D-dimer (alone) is useful in clinical practise. It is sensitive. But it is not spesific. Please correct the sentence. For example; "D-dimer is sensitive but not spesific marker for diagnosing PE/DVT and DIC". There are also more accurate references for this topic. For example; https://www.sciencedirect.com/science/article/pii/S0735109717397590?via%3Dihub

Response 5 This sentence was changed in accordance with the reviewer’s comment and reference 15 was changed to “Weitz JI, Fredenburgh JC, Eikelboom JW: A Test in Context: D-Dimer. JACC 2017; 70: 2411-20”

Reviewer 2 Report

The paper is well written, and the results are properly presented. The novelty of the paper is limited due to a recent publication by Zhang (DOI:10.1161/STROKEAHA.118.022563), however, the authors of this study aim to differentiate between different types of ACI, which has not been addressed in the other study. Moreover, the authors were able to improve the significance by combining sCLEC-2 with D-dimers.

Major points:

  • The authors suggest different treatments between cardioembolic and atherosclerotic ACI – what about acute thrombolysis?
  • Please provide more information about the patient cohort (have patients with anti-platelet medication been excluded?) – at which time point was the blood sample taken? How was the plasma/serum(?) prepared? Which anticoagulant? When studying platelet activation markers correct plasma preparation is critical.
  • Authors should comment on the platelet-specificity of sCLEC-2 and should discuss the difference to the classical platelet activation marker PF4 – what is the advantage?
  • Is the measured sCLEC-2 derived via membrane shedding or from microvesicles?
  • Please provide a data table to the ACI subpopulations – does sClec-2 correlate with platelet count?
  • Please include bars for healthy control in Figure 1
  • The additional value of Figure 3 and its implications should be explained/discussed in more detail
  • What do the numbers in Fig.2c and 3c mean?

Author Response

We thank you for your helpful and valuable comments on our study titled, “Soluble C-type lectin-like receptor 2 elevation in patients with acute cerebral infarction.” (Ref.: Manuscript Number: jcm-1305814). We have responded to all suggestions from the reviewers and have revised the manuscript accordingly. Our corrections are indicated as red letters with yellow highlight in the revised manuscript.

Respectfully yours,

Hideo Wada

Comment 6

The paper is well written, and the results are properly presented. The novelty of the paper is limited due to a recent publication by Zhang (DOI:10.1161/STROKEAHA.118.022563), however, the authors of this study aim to differentiate between different types of ACI, which has not been addressed in the other study. Moreover, the authors were able to improve the significance by combining sCLEC-2 with D-dimers.

Response 6 “Zhang X, Zhang W, Wu X, Li H, Zhang C, Huang Z, Shi R, You T, Shi J, Cao Y.: Prognostic significance of plasma CLEC-2 (C-Type Lectin Like Receptor 2) in patients with acute ischemic stroke. Stroke 2019; 50: 45-52” was cited.

 Major points:

Comment 7

  • The authors suggest different treatments between cardioembolic and atherosclerotic ACI – what about acute thrombolysis?

Response 7 Acute thrombolysis has a limitation of “within 4 hours” after the onset of ACI. We mainly treat cardioembolic ACI patients with anticoagulant therapy and atherosclerotic ACI patients with antiplatelet therapy. We have explain this difference in the treatments of cardioembolic ACI and atherosclerotic ACI in the Discussion.

 Comment 8

  • Please provide more information about the patient cohort (have patients with anti-platelet medication been excluded?) – at which time point was the blood sample taken? How was the plasma/serum(?) prepared? Which anticoagulant? When studying platelet activation markers correct plasma preparation is critical.

Response 8 Blood samples were obtained at admission and before treatment antiplatelet or anticoagulant therapy. Plasma was obtained after centrifugation two times (x 3,000 g). This sentence was added to the Materials and methods section.

 Comment 9

  • Authors should comment on the platelet-specificity of sCLEC-2 and should discuss the difference to the classical platelet activation marker PF4 – what is the advantage?

Response 9 CLEC-2 protein is highly and almost specifically and highly expressed in platelets and megakaryocytes in humans, but it is also expressed at lower levels in liver Kupffer cells [1]. Platelet factor 4 (PF4) and β-thromboglobulin (β-TG) are contained in α-granules and are used as classical platelet activation markers, while CLEC-2 is a platelet membrane protein. sCLEC-2 has several advantages over the classical platelet activation markers. PF4 and β-TG are easily released upon the minimal platelet activation that occurs during sampling and require plasma mixed with citrate, adenosine, theophylline and adenosine (CTAD); however, the effects of anti-coagulants on the sCLEC-2 ELISA were negligible. Moreover, while special techniques are required for blood sampling and sample preparation for PF4 and β-TG measurement, the standard blood collection procedures used in daily clinical laboratory tests have shown to be sufficient for sCLEC-2 measurement [2].

[1] Suzuki-Inoue K, Osada M, Ozaki Y. Physiologic and pathophysiologic roles of interaction between C-type lectin-like receptor 2 and podoplanin: partners from in utero to adulthood. Journal of thrombosis and haemostasis : JTH 2017;15:219-29.

[2] Kazama F, Nakamura J, Osada M, Inoue O, Oosawa M, Tamura S, Tsukiji N, Aida K, Kawaguchi A, Takizawa S, Kaneshige M, Tanaka S, Suzuki-Inoue K, Ozaki Y. Measurement of soluble C-type lectin-like receptor 2 in human plasma. Platelets 2015;26:711-9.

Comment 10

  • Is the measured sCLEC-2 derived via membrane shedding or from microvesicles?

Response 10 sCLEC-2 is derived via membrane shedding and via microvesicles. We have shown that in the ELISA system to measure sCLEC-2, both forms of sCLEC-2 are detected using 5D11 and 11E6 [2]. Since the chemiluminescent enzyme immunoassay (CLEIA) uses the same set of monoclonal anti-CLEC-2 antibodies, we assume that the measured sCLEC-2 is derived both via membrane shedding and from microvesicles.

[2] Kazama F, Nakamura J, Osada M, Inoue O, Oosawa M, Tamura S, Tsukiji N, Aida K, Kawaguchi A, Takizawa S, Kaneshige M, Tanaka S, Suzuki-Inoue K, Ozaki Y. Measurement of soluble C-type lectin-like receptor 2 in human plasma. Platelets 2015;26:711-9.

 Comment 11

  • Please provide a data table to the ACI subpopulations – does sClec-2 correlate with platelet count?

Response 11 The Data for the ACI subpopulations were added to Table 1. The relationship between sCLEC-2 and platelet counts were described in the Results section.

 Comment 12

  • Please include bars for healthy control in Figure 1

Response 12 Figure 1 was revised.

Comment 13

  • The additional value of Figure 3 and its implications should be explained/discussed in more detail

Response 13 The Discussion was revised in accordance with the reviewer’s comment.

Comment 13

  • What do the numbers in Fig.2c and 3c mean?

Response 13 This number was the sCLEC-2/D-dimer ratio ≥1200. The explain was added in legends.

Round 2

Reviewer 2 Report

We appreciate the improvement of the manuscript, however, still have some minor comments, which should be addressed: 

- Please correct the platelet counts in Table 1 – there seems to be a major error. Please provide statistics/significances for the added data.

- The anticoagulant for plasma preparation is still missing. Have all patients been anticoagulated the same?

- Figure 2c: the “c” needs to be at the correct place

- Figure 3 is still not properly explained in the results. Please explain the combinations

Author Response

Comment 1

- Please correct the platelet counts in Table 1 – there seems to be a major error. Please provide statistics/significances for the added data.

【Response 1】    Platelet counts were corrected and the statistics/significances were added.

Comment 2

- The anticoagulant for plasma preparation is still missing. Have all patients been anticoagulated the same?

【Response 2】Blood samples (1.8 ml) were obtained in plastic tube containing 0.2 ml of 3.2% sodium citrate buffer at admission and before antiplatelet or anticoagulant therapy. Plasma was obtained after the centrifugation of blood sample two times (3,000 x g).

Comment 3

- Figure 2c: the “c” needs to be at the correct place

【Response 3】 The “c” is moved to the correct place

Comment 4

- Figure 3 is still not properly explained in the results. Please explain the combinations

【Response 4】Results for Figure 3 were revised.
